# Southern Ocean BGC-Argo Detect Under Ice Phytoplankton Growth Before Sea Ice Retreat

Mark Hague[1] and Marcello Vichi[1,2]

[1]Department of Oceanography, University of Cape Town, South Africa
[2]Marine Research Institute, University of Cape Town, South Africa

**Correspondence:** Mark Hague, mark.hague@alumni.uct.ac.za

**Abstract.** The seasonality of sea ice in the Southern Ocean has profound effects on the life cycle (phenology) of phytoplankton residing under the ice. The current literature investigating this relationship is primarily based on remote sensing, which often lacks data for half the year or more. One prominent hypothesis holds that following ice retreat in spring, buoyant melt waters enhance available irradiance, triggering a bloom which follows the ice edge. However, an analysis of BGC-Argo data sampling under Antarctic sea ice suggests that this is not necessarily the case. Rather than precipitating rapid accumulation, we show that melt waters enhance growth in an already highly active phytoplankton population. Blooms observed in the wake of the receding ice edge can then be understood as the emergence of a growth process that started earlier under sea ice. Indeed, we estimate that growth initiation occurs, on average, 4-5 weeks before ice retreat, typically starting in August and September. Novel techniques using on-board data to detect the timing of ice melt were used. Furthermore, such growth is shown to occur under conditions of substantial ice cover (>90% satellite ice concentration) and deep mixed layers (>100 m), conditions previously thought to be inimical to growth. This led to the development of several box model experiments (with varying vertical depth) in which we sought to investigate the mechanisms responsible for such early growth. The results of theses experiments suggest that a combination of higher light transfer (penetration) through sea ice cover and extreme low light adaptation by phytoplankton can account for the observed phenology.

## 1 Introduction

The annual advance and retreat of Antarctic sea ice is the largest seasonal event on Earth, covering some 15 million km$^2$ (Massom and Stammerjohn, 2010). Such considerable seasonal changes have profound effects on the life cycle (phenology) of phytoplankton residing under the ice (Sallée et al., 2015; Ardyna et al., 2017; Hague and Vichi, 2018). However, the exact character of such effects is currently unknown, primarily because studies investigating phenology in these regions have relied on satellite data, which can contain missing data for half the year or more (Cole et al., 2012; Racault et al., 2012). In particular, the winter and early spring periods are not taken into account, despite the important role they play in both the overall phenology and subsequent summer production (Llort et al., 2015; Ardyna et al., 2017; Boyce et al., 2017).

Here we present the first ever comprehensive characterization of under ice phenology for this period. This is achieved by leveraging under ice data collected by ARGO profiling floats equipped with a suite of biogeochemical sensors, deployed as

part of the SOCCOM project (https://soccom.princeton.edu/). Of primary interest to us here are the mechanisms controlling the timing of phytoplankton growth initiation in the unique environment of the Seasonal Sea Ice Zone (the SSIZ, defined as the ocean region seasonally covered by sea ice). A dominant idea in the literature with regards to such mechanisms holds that following ice retreat in spring, buoyant melt waters tend to enhance irradiance levels by rapidly shoaling the mixed layer. This alleviation of light limitation (coupled perhaps with nutrient input) is then used to explain why blooms are often observed in the wake of the receding ice edge (Smith and Nelson, 1985; Smith and Comiso, 2008; Briggs et al., 2017; Sokolov, 2008). The implication here is then that, prior to the release of melt waters, growth rates remain low, only increasing substantially in response to melting. Hence, a prediction of the hypothesis (which we may term the "melt water hypothesis") is that the timing of melting should precede the timing of rapid growth. This is a somewhat subtle point, since the relevance of melt waters is usually brought up to explain the presence of blooms, and so is often not explicitly linked to phenology (e.g. Taylor et al. (2013); Uchida et al. (2019)). Nevertheless, the hypothesis implicitly assumes that phenology is strongly affected by the release of melt water.

However, there is increasing evidence that this is not necessarily the case. In an early study, Smetacek et al. (1992) documented an intense bloom under pack ice conditions in early spring (before melting) in the Weddell Sea ice shelf region. More recently in the Arctic a similarly intense phytoplankton bloom was observed in the Chukchi Sea under complete ice cover ranging from 0.8 to 1.3 m thick Arrigo et al. (2012). Although this bloom was observed in summer, Assmy et al. (2017) have recently shown that under ice blooms may develop even earlier in the Arctic due to the presence of leads in spring. In the Southern Ocean, evidence is emerging of earlier than expected growth in deep mixed layers within the SSIZ (Uchida et al., 2019; Prend et al., 2019). The important feature of these studies for the present discussion is that high growth rates have been observed prior to melting and under complete (or near-complete) ice cover. However, the present literature has left several issues related to under ice phenology unresolved.

Firstly, studies focus almost exclusively on spring and summer and hence miss any potential growth occurring in winter. Indeed, it is assumed that such growth is negligible even though this has not been explicitly shown. Second, much attention is paid to regions of high biomass (i.e blooms) and their associated environmental conditions. Although these regions are no doubt of great interest, their study does not necessarily contribute to an understanding of the mechanisms controlling phenology in general. This is especially true in the Southern Ocean where large spatial variability is common (Thomalla et al., 2011). Third, the bulk of present literature is based on studies of Arctic under ice phenology. Antarctic sea ice is distinct in being generally thinner and more dynamic, as well as having much more snow year-round that does not form melt ponds (Vancoppenolle et al., 2013). Consequently, the presence of Marginal Ice Zone (MIZ) conditions is also much more prevalent in the Southern Ocean. Note that we would define the MIZ here by dynamical considerations such as wave propagation (i.e. the MIZ may be defined as the region where wave attenuation is below a given threshold) and not a satellite ice concentration threshold (for example, see Squire (2007); Meylan et al. (2014)). This means that seasonal variations of light and nutrients are likely very different, motivating special attention to this unique environment.

## 2 Methods

The work presented here employs several novel techniques to detect the timing of phenological events in the seasonally ice covered Southern Ocean. In particular, the use of profiling float data to detect both the timing of melt and growth initiation avoids several of the shortcomings inherent to satellite and ship-based studies which characterize the present literature on under ice phenology. First, if in-situ data are used, they are limited in space and time and are usually compared to satellite products as a consequence. However, direct comparisons of this kind are often associated with large uncertainties stemming from the coarse spatial resolution of satellite products, as well as differences in the measurement technique used to produce in-situ and satellite data. By using a consistent observing platform we largely overcome these issues, while still achieving good spatial and temporal coverage at seasonal time scales (see Section 3.1). Another clear advantage of the SOCCOM dataset is the availability of depth information, allowing us to simultaneously compare the seasonal evolution of temperature, salinity and chlorophyll-a (chl-a) in the water column, and compare this to diagnostics such as the mixed layer depth (MLD). Unfortunately, floats analysed in this study are not equipped to measure Photosynthetically Active Radiation (PAR) under ice.

We now move on to a more detailed discussion of the methodology used to detect melt water release and growth initiation. Following this we will describe the biogeochemical model experiments used to investigate the drivers of under ice growth.

### 2.1 Data Sources

Float data used in this study is made available by the SOCCOM project, which can be downloaded through their website (https://soccompu.princeton.edu/www/index.html). Analysis was done on data from 2014 - 2019, making use of chl-a, pressure, temperature, salinity and position data available at a 10-day frequency. Satellite sea ice concentration for the period January 2015 to April 2019 is taken from the NOAA/NSICD Climate Data Record (version 3), which makes use of 2 passive microwave radiometers: the Special Sensor Microwave Imager (SSM/I) and the Special Sensor Microwave Imager/Sounder (SSMIS). The data are downloaded at daily resolution on the NSIDC polar stereographic grid with 25 x 25 km grid cells. Finally, incident solar radiation at sea level used to force the model simulations is taken from European Centre for Medium-Range Weather Forecasts (ECMWF) ERA-Interim reanalysis dataset. The data resolution is daily (the mean of the day-night cycle) on a 0.75°x 0.75°regular grid.

### 2.2 Detection of Phenological Events

**Under Ice Detection**

The first step in our investigation of under ice phenology was to determine which profiles could be classified as sampling under ice. Since many of floats deployed in the SOCCOM project are intended to sample under ice, an ice avoidance algorithm is utilized onboard. The ice-sensing algorithm simply compares the median temperature between ∼50 m and 20 m during ascent to a threshold temperature of -1.78°C. If the observed value is lower than the threshold, it is assumed that sea ice is present overhead. The float then terminates its ascent, stores the profile data and returns to its parking depth (Riser et al., 2018).

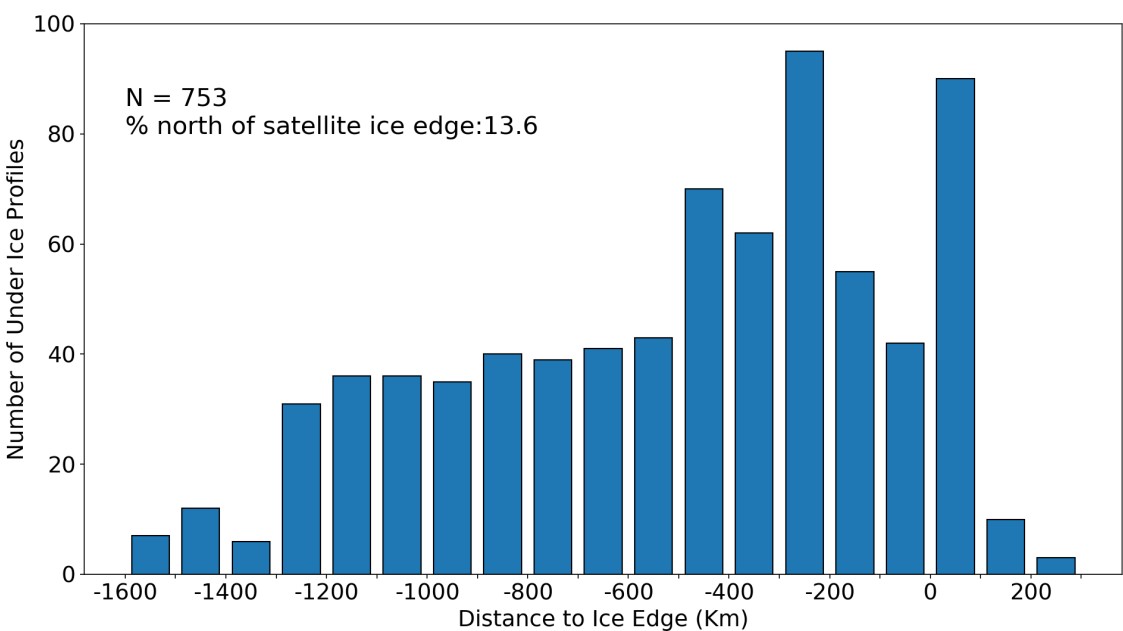

**Figure 1.** Distribution of great-circle distances of under ice profiles to the estimated satellite sea ice edge (latitude of 15% sea ice concentration contour). Negative values indicate that the profile is poleward of the ice edge.

Since the freezing point of sea water depends both on temperature and salinity, we chose to include near-surface salinity measurements in our revised under ice detection algorithm. That is, for each profile the freezing temperature, based on the salinity closest to the surface, is computed and compared to the temperature measured at the same depth. It is important to note that the depth of these near-surface measurements will vary from $\sim$20 - 25 m in winter, to $\sim$0 - 5 m in summer. This is because of the on-board ice avoidance algorithm described above: in winter the temperature threshold is generally exceeded, and so sampling ceases $\sim$20 m from the surface, while in other months this condition is generally not met and so floats are able to sample much closer to the surface. Therefore, since the on-board ice avoidance algorithm is intentionally conservative, it may assume there is ice present when in fact melting has already occurred (or when the float has moved out of an ice covered region). Conversely, the ice avoidance algorithm may also incorrectly determine that melting has occurred and attempt to surface when ice is still present overhead.

While the fact that winter profiles generally only sample up to $\sim$20 m may seem unimportant, it actually has significant bearing on the under ice detection algorithm used in this study. This is because within the upper 20 m in winter, water is generally above its freezing point (if salinity is taken into account). Therefore, in order to delineate under ice from open ocean profiles, one has to assume some degree of cooling from the last measurement in the profile to the surface. Since the realized degree of cooling over this winter surface layer cannot be observed, we tested several values (the corresponding effect they

had on a key result of the paper is shown in Figure 5). The orange and green curves in this Figure depict the change in the
probability density function when increasing and decreasing the cooling threshold (rc) by 20%, respectively (the details of
what is depicted in the figure is discussed below in the "Growth Initiation" section). The blue curve and associated histogram
depicts the chosen value of 0.1°C used in this study (i.e. we assume a decrease of 0.1°C from ~20 m to the surface). We would
note that the essential features of the distribution remain unchanged in this sensitivity test.

In addition to the above testing, two further checks were performed to assess the validity of using an assumed rate of cooling
to detect under ice profiles. The first approach is shown in Figure 1, which plots the distribution of distances of under ice
profiles to the satellite ice edge. Here the ice edge is defined by the 15% sea ice concentration contour, following previous
satellite-based studies (e.g. Stroeve et al. (2016)) We found that the vast majority of profiles where located 100 km or more
south of the ice edge, with 13.6% being north of the edge. It is important to point out here that while sampling under ice, floats
do not communicate their location, since they are prevented from surfacing. A simple linear interpolation is used to estimate
the location of the under ice profiles (based on the relative time stamp difference), with an approximate maximum error of 100
km as reported by Riser et al. (2018). It is precisely because of this uncertainty that we chose to use on-board data to detect
under ice profiles (as well as to detect melting), as opposed to flagging profiles as under ice based on their relative position to
the satellite ice edge. The distribution shown in Figure 1 is included to illustrate that there is broad agreement between the two
methods, although the use of on-board data should be more accurate given the uncertainty of the float location.

The second approach used to assess the under ice detection method involved visual inspection of time series of mean mixed
layer temperature and salinity like those shown in Figure 2A (a discussion of how MLD is defined is given below under
"Melt Onset Detection"). This consisted of comparing the timing of the transition from under ice to open ocean (depicted by
the black vertical line in Figure 2A), with the associated changes in temperature and salinity. Both raw (dashed) and filtered
(solid) time series are shown, with a first-order, low-pass Butterworth digital filter employed with a cut-off frequency of 0.1
Hz. By inspecting a subset of floats sampling under ice, we found good visual agreement between the computed timing of
the transition (black vertical line) and the corresponding tendency of the curves toward freshening and warming of the surface
ocean. The best agreement is achieved by assuming a relative temperature difference (between the last winter measurement
and the surface) 0.1°C as described above, which is why this value was chosen over other candidates. A sample of time series
for floats other than that shown in Figure 2A is provided in supplementary Figures S1-S4.

**Melt Onset Detection**

Once a transition from ice cover to open ocean has been established, our algorithm then verifies that these changes are associ-
ated with melting. This is done by computing time derivatives of surface temperature and salinity at the time of transition (data
are taken from measurements closest to the surface). In order to be classified as a melt event, the temperature derivative must
be positive (i.e increasing temperature) with a negative salinity derivative that is persistent for 1 month following transition.
An example of a such a melt event is shown in Figure 2A, where salinity (blue lines) decreases gradually for ~1 month prior
to transition, while temperature (red lines) begins to steadily increase after remaining consistently below freezing. At least
three consecutive under ice profiles (equivalent to ~1 month since profiles are at 10-day frequency) are needed to detect a

melt event. In cases where multiple transitions occur in one season, the transition with the strongest signal (i.e steepest time derivative) of warming and freshening is chosen. This enables us to filter out transitions which occur as a result of advection or high frequency warming associated with synoptic variability.

Apart from the three criteria discussed above (transition from under ice to open ocean, positive temperature derivative, negative salinity derivative), additional inspection of time series of stratification depth (our chosen metric for assessing vertical mixing, termed $N_d$) was performed. This depth is defined as the point at which the Brunt-Väisälä frequency reaches its maximum value in the upper water column, implying a region of maximum resistance to mixing (Gill, 1982). Furthermore, this measure of the depth of the mixed layer has been shown to be more ecologically relevant in the Southern Ocean than other more traditional methods involving density/temperature thresholds (Carvalho et al., 2017).

As is discussed in Section 1, the release of melt waters tends to stratify the surface ocean, and so $N_d$ should rapidly decrease following the detected melt event. In Figure 2B we show an example of such a time series of $N_d$ (in blue), with profiles flagged as under ice shown with red stars. One can clearly see that $N_d$ shoals rapidly at the point of transition from under ice to open ocean, providing further confidence in the melt detection algorithm. Additional figures which were used to verify the algorithm are provided in supplementary Figures S1-S4, which show similar results.

**Growth Initiation (GI)**

Our main metric for assessing the relationship between melting and phenology is termed growth initiation (GI). It is defined here as the point at which the time derivative of mean mixed layer chl-a exceeds the median time derivative computed for the growth period in question. These time derivatives, here taken as a proxy for growth rates, are only computed over the period of positive growth. This period is determined from a filtered time series of mean mixed layer chl-a used to remove variability at the 10-day sampling frequency (the actual value of the median is computed from the raw signal). A first-order, low-pass Butterworth digital filter is employed with a cut-off frequency of 0.1 Hz. An example of the resulting filtered time series is shown in Figure 2B and compared to the original raw signal. Also shown in the figure by the black vertical line is the timing of GI for this particular season. The distance between the 2 black vertical lines in panels A and B in Figure 2 then denotes the timing difference between melting and GI as shown in Figure 5 for all float data.

Following Racault et al. (2012), early stages of growth are usually quantified using a metric termed bloom initiation, which is defined as the time at which chl-a concentration first exceeds the long term median plus 5%. However, this method is unsuitable in this study for several reasons. First, our time series are, at most, 4 years long, and on average only 2 years long, precluding an estimation of any long term threshold value. Second, our focus is on the conditions which trigger *growth*, not necessarily a *bloom*, which again implies that a comparison to some longer term value must be made. Finally, we believe a metric based on growth rates (as opposed to an absolute threshold value) to be more appropriate, since it avoids any biases in the median which may be created by long periods of close to zero chl-a concentration under ice (followed by a rapid increase).

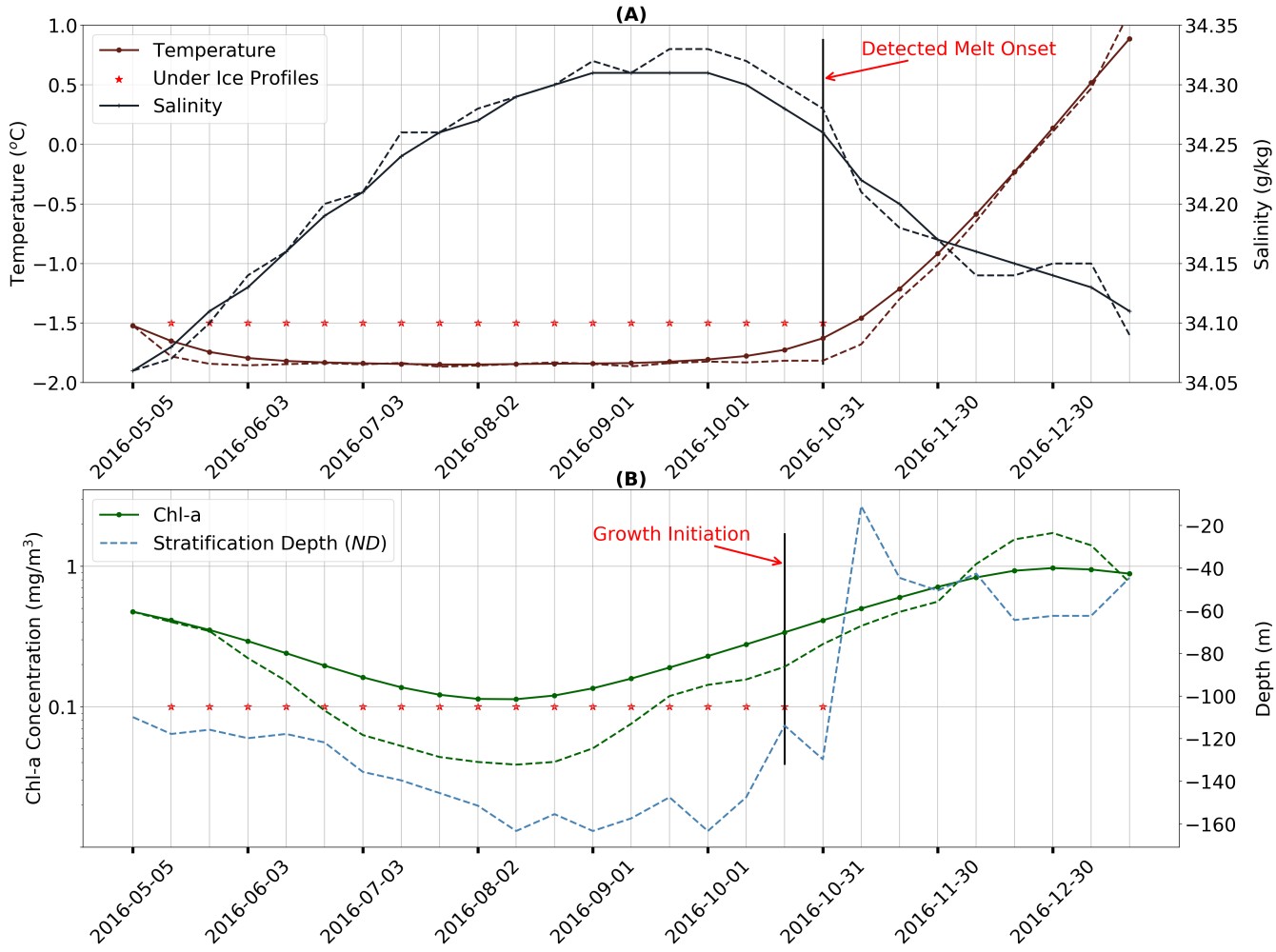

**Figure 2.** Time series of key properties illustrating the methodology used for melt and growth detection. (A): Near surface temperature (dark red) and salinity (grey) from May to December 2016 in the Ross Sea sector ( 65°S). Red stars indicate profiles flagged as under ice. Solid lines with markers are filtered time series (higher frequencies have been removed), dashed are raw data. (B): Mean mixed layer chlorophyll-a (dark green) and mixed layer depth plotted for the same period as in (A). For more information refer to section 2.2.

### 2.3    Model Experiments

#### 170    2.3.1    Model Set-up

A biogeochemical box model is employed in this study to investigate the drivers of under ice growth. The model is based on the Biogeochemical Flux Model (BFM) framework, for which documentation can be found in Vichi et al. (2015). Our particular configuration is a "0.5D" box model where all the major components of the marine biogeochemical system are simulated, namely, phytoplankton, zooplankton, organic and inorganic matter, nutrients and bacterioplankton. The model is
termed "0.5D" due to the fact that the depth of the box is able to vary to simulate the effect of vertical mixing. In this case the only effect taken into account is the attenuation of light with increasing mixed layer depth.

Since our study is process-oriented, we chose to simplify the model as much as possible, while still retaining the major features of interest. Accordingly, only 1 phytoplankton (diatoms) and 2 zooplankton groups (omnivorous mesozooplankton and heterotrophic nanoflagellates) are simulated. In terms of nutrients, phosphate, nitrate and ammonium are included, as well
as silicate and iron. Initial nutrient conditions where chosen to be representative of the Southern Ocean south of $\sim$60°S, with non-limiting concentrations of nitrate (31.8 $mmol/m^3$), phosphate (2 $mmol/m^3$) and silicate (40 $mmol/m^3$) (Sarmiento and Gruber, 2006). An initial dissolved iron concentration of 0.3 $\mu mol/m^3$ (Tagliabue et al., 2014) is applied to all experiments, which gave the most realistic magnitude of summer growth when compared to float data.

The model is forced daily with solar radiation, satellite sea ice concentration, float temperature and salinity, as well as mixed
layer depth derived from float data (refer to Section 2.1 for data sources). Time series of incident PAR and mixed layer depth (as estimated by the stratification depth, $N_d$) for each study region is provided in Figure S5. Light available at the surface is scaled by the sea ice concentration by simply multiplying the incident radiation by the percentage of open ocean derived from remote sensing data. Note that the Weddell Sea study regions (W60 and W65), using an analytical light forcing described in Vichi et al. (2015).

#### 190    2.3.2    Experiment Design

Three core experiments were conducted in 4 study regions, with each run having a spin-up time of 10 years to allow for adjustment to a repeating annual cycle (although in most cases adjustment took only a few years). In Table 1 we provide an overview of the available float data in each study region. For every complete time series of float observations we performed the set of three core experiments. First, 2 sets of experiments were run to test the effect of sea ice cover on phytoplankton
phenology: a run with no ice (OPEN) and a run with imposed satellite sea ice concentration (ICE). A third experiment sought to test the combined effect sea ice cover and increased low light adaptation by phytoplankton had on phenology (LLA). This was achieved by increasing the initial slope of the Photosynthesis-irradiance curve by a factor of 10, thus enhancing photosynthetic efficiency at light levels close to zero. This value is equivalent to what is commonly used for sea ice algae (Tedesco et al., 2010).

**Table 1.** Number of floats sampling in each year for the 4 study regions. This number then corresponds to the number of model runs done in each region for each of the three core experiments discussed in the text. W60 = Weddell Sea region at ~60°S; W65 = Weddell Sea ~65°S; B70 = Bellingshausen/Amundsen Sea ~70°S; R75 = Ross Sea south of 75°S.

| | **2015** | **2016** | **2017** | **2018** | Float ID |
|---|---|---|---|---|---|
| **W60** | 1 | 1 | 1 | 1 | 5904397 |
| **W65** | 2 | 2 | 2 | 2 | 5904468; 5904471 |
| **B70** | 0 | 0 | 2 | 2 | 5904859; 5905075; 5905080 |
| **R75** | 0 | 0 | 3 | 2 | 5904858; 5904857; 5904860 |

## 3 Results

The results presented here fall under two general themes. In the first section we will test the melt water hypothesis outlined in Section 1, by comparing the timing of growth initiation (GI) with that of sea ice retreat. Following this we will present results from a set of simple model experiments in an attempt to explain for the observed phenology. In these experiments we investigate the role sea ice cover and phytoplankton low-light adaptation play in controlling winter/spring growth. By placing the experiments in 4 distinct study regions with different physical conditions, we also utilize the spatial and temporal variability available in the float dataset to derive results of wider regional applicability.

### 3.1 Observed Under Ice Growth

Figure 3 plots the approximate mean location of the 42 melt events captured in the BGC-Argo dataset. From this map it is clear that a fairly broad spatial distribution is achieved, with all the major ocean basins sampled. However, the Atlantic and Pacific oceans are better represented, with the Weddell, Bellingshausen and Ross Seas having the highest concentration of sampling. Meridionally floats sample between approximately 60 and 70°S, and cover the period 2015 to 2018. Based on this spatial and temporal distribution, we can expect a large variability in oceanographic conditions. This in turn leads to the large spread observed in the timing of GI (from September to January), represented by the colours of the points in Figure 3. While there is some indication of the expected progression towards later GI as one moves south (lighter colours), large interannual variability is observed where points are clustered together in space but nevertheless have very different GI values (this difference is as large as 8 weeks in Weddell Sea at ~65 °S).

In Figure 4A we show more explicitly the relationship between growth initiation timing and latitude. Here we find a statistically significant correlation (p = 0.001) of -0.47, implying that 22% of the variance in GI may be explained by variability in latitude alone. Conversely, the relationship between GI and the timing of melt water release is insignificant at the 5% level (p = 0.07) with a lower correlation of 0.27 (Figure 4B). Furthermore, almost all events fall below the 1:1 line in Figure 4B, revealing that GI tends to precede the release of melt waters.

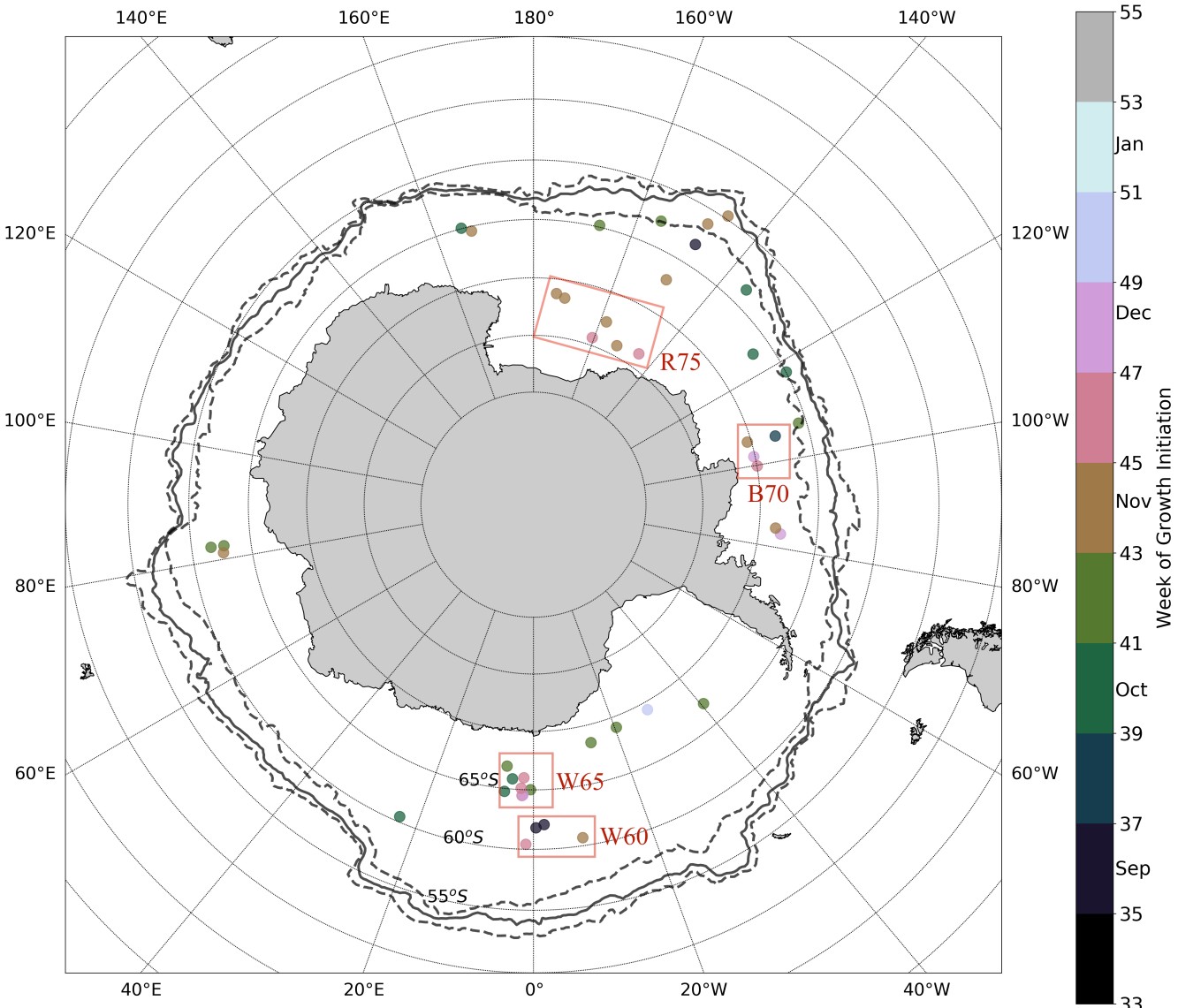

**Figure 3.** Approximate locations of melt events identified in the under ice BGC-Argo dataset. The solid black line represents the mean maximum extent of the 15% sea ice concentration contour for the period 2015 - 2018, while dashed lines represent the interannual variability. The colour of each point represents the timing of growth initiation (GI) in weeks of the year. Red boxes refer to study regions discussed in the text.

Consequently, in Figure 5 we plot the distribution of the difference in timing between GI and melting. For the majority of the observed events, GI occurs well before the release of melt waters (the mean timing difference is 4.5 weeks). Furthermore, for 35% of the events, GI is observed more than 35 days before melting, with a further 25% preceding melting by 25-35 days.

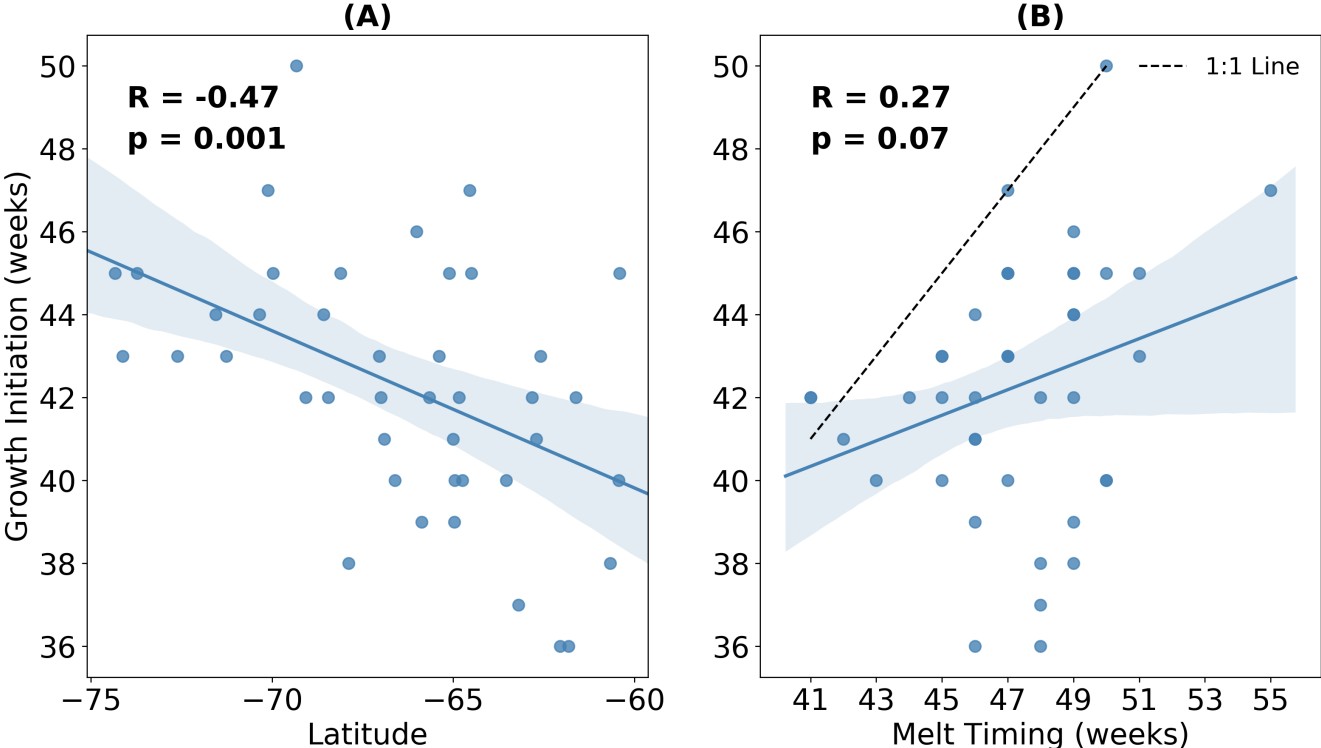

**Figure 4.** Timing of growth initiation (GI) plotted against **(A)**: average latitude and **(B)** : timing of sea ice melt for each of the 42 melt events shown in Figure 3. Overlain in blue is the linear regression with the 95% confidence intervals for 1000 bootstrapped resamples shaded in light blue. On panel B the dashed black line represents the 1:1 line.

Only 10%, or 4 events, occur either at the same time as or after sea ice retreat. In a complementary analysis, we included chl-a data from $\sim 50$m below the estimated mixed layer depth in the calculation of GI. Overall this tended shift GI even earlier in the year by diluting the summer concentration, although in 4 cases significant chl-a below the mixed layer enhanced the spring growth rate and thus delayed GI.

We would note that our definition of GI (detailed in section 2.2) is likely to be more conservative than methods employing a threshold value (i.e likely to delay growth initiation). In addition, GI was also computed using the mean mixed layer particulate organic carbon as opposed to chl-a, which resulted in a timing difference distribution similar to that shown in Figure 5 (albeit with a smaller time difference, see Supplementary Figure S8). In terms of vertical mixing, average stratification depth ($N_d$) at GI is $\sim 128$ m, with a standard deviation of 51 m. In Figure S6 we show the value of $N_d$ at GI for all 42 events, as well as the relationship between $N_d$ and GI. We found that $N_d$ generally ranged between $\sim 75$ and $\sim 160$ m at the timing of growth initiation, with no correlation between $N_d$ and GI. Table 2 highlights some of the salient properties of the data set investigated in this study, as well as summarizing the major findings discussed above.

**Table 2.** Summary of properties of under ice BGC-Argo dataset. GI = Growth Initiation.

| | |
|---|---|
| Total Floats | 99 |
| N Floats Under Ice | 20 |
| N Profiles Under Ice | 753 |
| N Melt Events | 42 |
| Mean Time Series Length | 27 months |
| Mean Timing of GI | Week 42 (Mid-October) |
| Mean Timing of Melt Onset | Week 49 (Early December) |
| Mean Chl-a at GI | 0.14 mg/m$^3$ |
| Mean Peak Chl-a | 2.31 mg/m$^3$ |
| Mean Stratification Depth (MLD) at GI | 128 m |

While the results discussed up to this point incorporate data from all available under ice floats, in Figure 6 we focus on 3 floats which sampled in close proximity to each other in the Ross Sea. In the figure, each bold line plots the mean value of 5 time series which correspond to different melt events. Events are separated in space and time; in this particular case 2017 and 2018 were sampled by 3 floats, which resulted in 5 time series (2 each, with one of the floats only sampling in 2017). This allows for a clear comparison of the seasonality of chl-a and sea ice, serving as a good example of how phytoplankton are able to sustain growth under near complete (according to satellite information) ice cover. Indeed, in this particular case average satellite sea ice concentrations were consistently above 90% until late November, by which point chl-a has already been steadily increasing for 2-3 months. Examples of other regions can be found in Figures S7.

## 3.2 Regional Modelling of Under Ice Growth

In order to further investigate which factors may drive early growth under ice, we conducted several simplified model experiments in 4 study regions. The objective here is to determine which experiments most closely resemble the observed seasonality of mixed layer chl-a, thereby inferring which factors may be important in promoting under ice growth. Each region was chosen based on the spatial distribution of melt events shown in Figure 3. In the Weddell Sea close to 0°, 2 distinct clusters of melt events are seen in Figure 3, one centred just south of 60°S, the other around 65°S. For ease of identification we call these regions W60 and W65, respectively. In the Ross Sea we selected 3 floats which sampled in relative proximity north of 75°S (region R75). Finally, we selected 3 additional floats which sampled in the Amundsen/Bellingshausen Sea (just north of Pine Island Bay) around 70°S (region B70). This allowed us to compare experiments run under different forcing conditions (in particular, sea ice concentration, light and mixed layer depth).

Three core experiments were conducted for each region, consisting of first running with no sea ice forcing (OPEN), then with satellite derived ice concentration (ICE), and finally with the low light efficiency of phytoplankton enhanced by a factor of 10 (LLA; sea ice forcing is also kept for these runs). Within each of the four study regions, this set of experiments is conducted

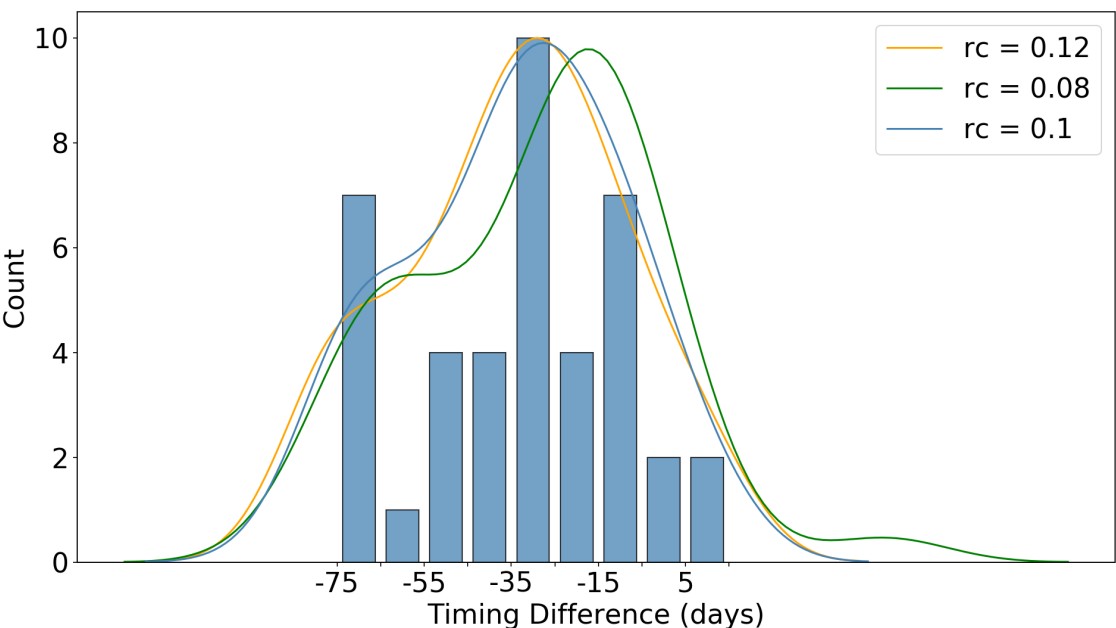

**Figure 5.** Distribution of the difference in timing (in days) between growth initiation (GI) and melt onset (for all floats sampling under ice). GI is defined as the point at which the time derivative of mean mixed layer chl-a exceeds the median time derivative (computed for the growth period). Negative values in the distribution indicate that GI has occurred prior to the detected melt onset. Curved lines represent the probability density functions for several values of the assumed cooling threshold (rc) in the upper ∼20 m of the water column. This value represents an assumed decrease in temperature over the upper ∼20 m, which is required to delineate under ice from open ocean profiles (since floats do not sample the upper ∼20 m in winter, they do not sample water below the freezing point). Refer to Section 2.2 for a discussion of the methodology used to produce the Figure.

for each year available in the float data (and in some cases multiple times for the same year if more than one float sampled the region; see table 1). Refer to section 2.3.1 for more information on the model setup and forcing, and to section 2.3.2 for
experiment design.

The results of all experiments are shown in Figure 7 and compared to the phenology obtained from float data. This is done by averaging each of the three core experiments across each study region to give the mean time series of mixed layer chl-a shown by the red (OPEN), blue (ICE) and green (LLA) bold curves (the same is done for the corresponding float observations shown in black).The shading around each of these curves in Figure 7 represents variability across the set of model runs (or float
time series), as was discussed above for Figure 6. For example, in the case of the B70 region (panel C), a total of 4 runs where conducted for each experiment (corresponding to the 4 float time series available, see table 1), and so the shaded regions show the variability present across 4 time series.

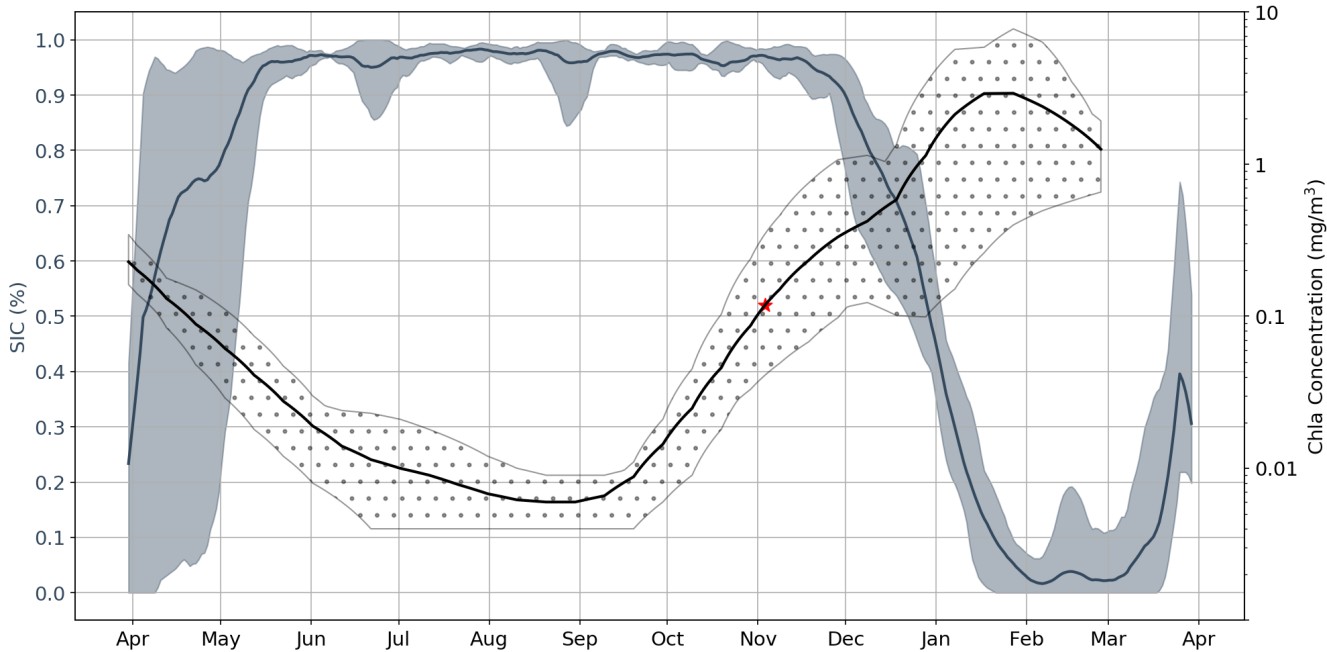

**Figure 6.** Satellite sea ice concentration (SIC) versus ARGO-float chl-a for the region R75. Shaded regions around each line represent both the spatial and temporal variability present in each dataset.That is, each bold line plots the mean value of 5 time series which are each associated with a specific melt event. Events are separated in space and time; in this particular case 2017 and 2018 were sampled by 3 floats (see table 1), which resulted in 5 time series (2 each, with one of the floats only sampling in 2017). The red star represents the mean value of GI.

By comparing key phenological features of the time series shown in Figure 7, we can examine which of the three model configurations most closely matches the float time series. The primary features of interest to us here are timing of initial growth (i.e. a switch declining to increasing chl-a concentrations) and the subsequent rate of growth in spring. Other features such as the timing of peak concentration and the intensity of seasonality (i.e. summer - winter chl-a concentration) are not discussed in detail here.

The key finding of the figure is that winter and spring phenology are most closely captured by LLA experiments in the Ross and Bellinghausen/Amundsen seas (regions R75 and B70, respectively), while in the Weddell Sea (regions W65 and W60) a combination of OPEN and LLA experiments can account for the phenology of this period. That is, in the Weddell Sea the timing of the transition from negative to positive derivative in chl-a is better represented by OPEN experiments, while the subsequent rate of growth in spring is more closely simulated by LLA experiments (panels A and B of Figure 7). Indeed, across all regions OPEN experiments seem to capture well the timing of the minimum chl-a concentration in winter, but then greatly over estimate the spring growth rate. In almost all cases, the ICE experiments overly dampened growth in winter and spring, with the switch from negative to positive mixed layer chl-a derivative occurring significantly later than observations. However,

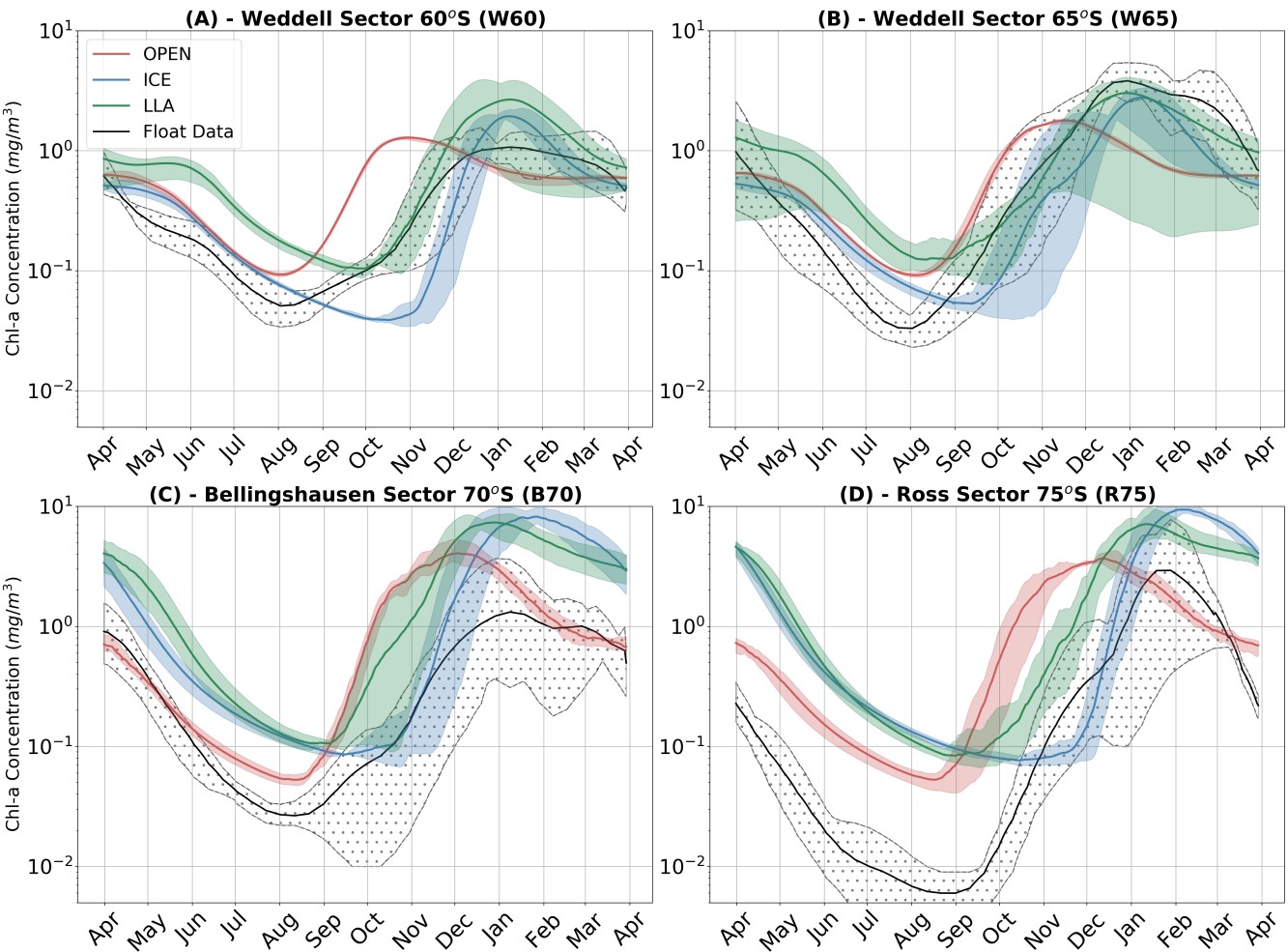

**Figure 7.** Time series of mean mixed layer chl-a for each of the 4 regions discussed in the text. In each panel the observed values (black) are compared to 3 model experiments; runs with no sea ice are shown in red (OPEN), runs with ice in blue (ICE), and runs with both ice and enhanced low light efficiency by phytoplankton are plotted in green (LLA). The shaded regions for each curve represent the spatial and temporal variability present in each dataset as in Figure 6. Note that the time series run from April to April.

in the Weddell Sea (W60 and W65) this model configuration suffers least from the compressed seasonality particularly evident in the winter months of other experiments.

In Figure 8 we show the timing of GI for each region and experiment, providing a more quantitative view of the relative changes in phenology (each point in the figure represents a separate year or location). GI for the model time series is computed in the same manner as in the float data (see section 2.2), although there was no need for filtering. While the LLA set of experiments generally performs best at reproducing GI, there are notable exceptions in each of the 4 study regions. In the W60 region the observed GI occurs between early September and mid-October, with OPEN experiments having growth too early

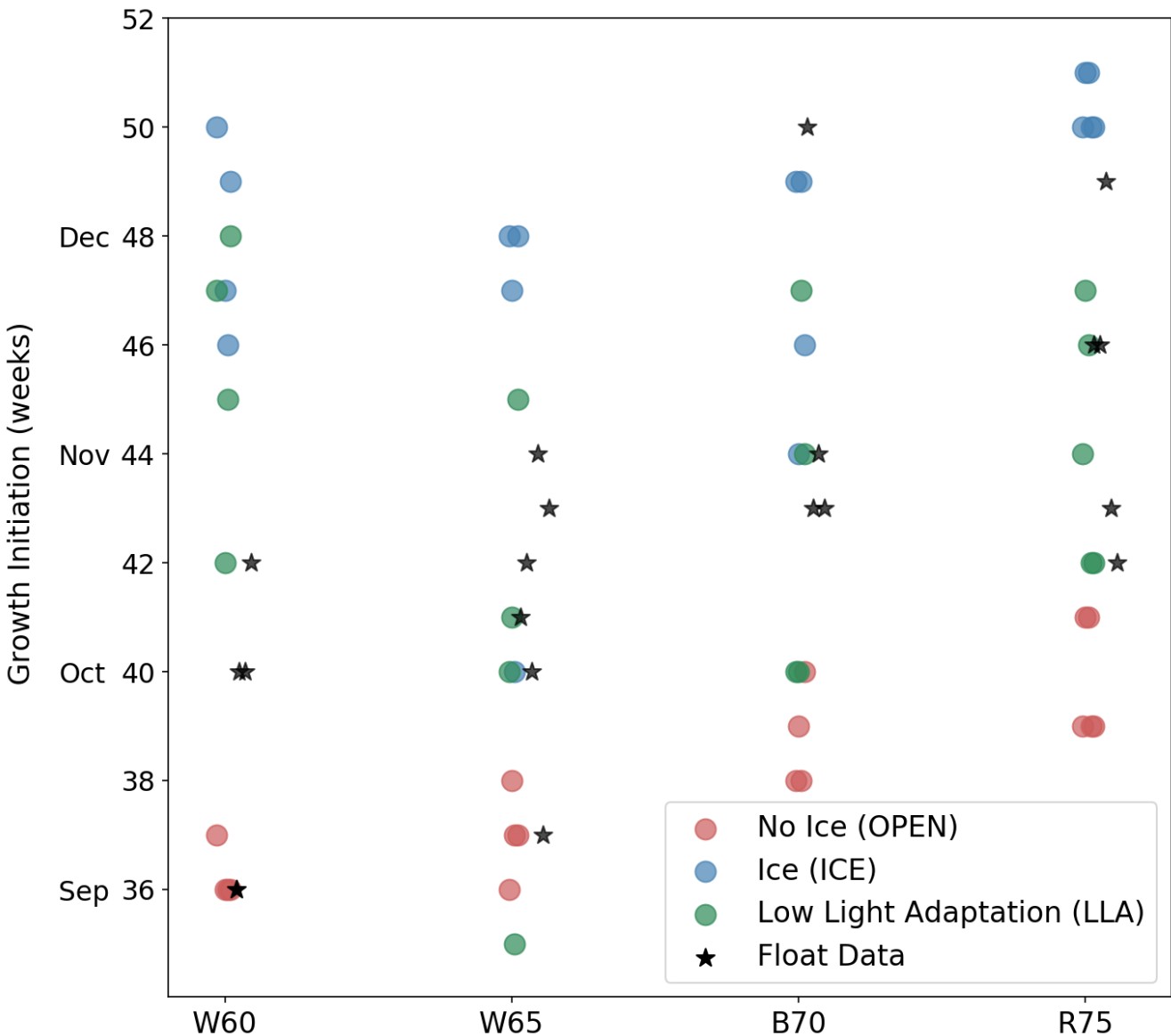

**Figure 8.** Timing of GI for each study region (horizontal axis) and model experiments (coloured points). Corresponding values from float data are indicated by black stars.

and LLA experiments too late. Moving further south to W65, we see that only LLA is able to capture the observed variability in GI, but that in some cases OPEN provides the best fit to data. Continuing south and west, both B70 and R75 contain cases where GI is best described by ICE simulations. In the following section we will bring together both the observational and

modelling results discussed thus far, thereby shedding light on the possible mechanisms leading to under ice growth in the Antarctic winter and spring.

## 4 Discussion

**Relationship between Melting and Growth**

The central question of the present study relates to what conditions are necessary to trigger phytoplankton growth in the Antarctic SSIZ. As has been outlined in Section 1, a popular hypothesis holds that the release of buoyant melt waters following sea ice retreat shoals the mixed layer, relieving light limitation and triggering rapid growth. In contrast to previous studies (e.g. Smith and Nelson (1985); Smith and Comiso (2008); Sokolov (2008); Taylor et al. (2013)) relying on satellite data or models, we were able to thoroughly test this hypothesis by utilizing a unique in-situ dataset of under ice profiles from BGC-ARGO

floats. In particular, we were able to test two predictions of the hypothesis; first, that at least part of the variability in the timing of growth initiation (GI) may be explained by the timing of sea ice melt, and second, that GI should either be synchronous with or occur after the release of melt waters.

   Based on the data analysed here, we do not find evidence which convincingly supports either claim. In Figures 5 and 6, we clearly demonstrate that phytoplankton are able sustain growth long before significant freshening of the surface ocean. It

is important to reiterate here that GI is based on the rate of growth exceeding the median rate, and so the tendency of GI to precede melting (as illustrated by the timing differences between these events shown in Figure 5) suggests that the rate of growth is already well above average prior to ice retreat. This explains why GI and melting are not correlated in time (Figure 4B); the release of melt waters does not appear to relieve light and/or nutrient limitation and so variability in melt timing cannot account for variability in GI. GI is instead correlated more strongly with latitude (Figure 4A), suggesting that phytoplankton

are responding to changing incident light conditions rather than fresh water fluxes. To be clear, the latitudes plotted in Figure 4A are computed based on the approximate location of the float at GI, which in almost all cases corresponds to an under ice condition. Therefore, the correlation found in this figure implies that light may be non-limiting under Antarctic sea ice (at least in the conditions sampled by the floats), provided it is late enough in the season for there to be sufficient light available at the surface.

Also noteworthy is the extent to which growth occurs prior to melting, with ∼60% of events preceding melting by a month or more. As is discussed in 3.1, our float dataset samples in a wide variety of environmental conditions, which exhibit very different sea ice and vertical mixing regimes. This suggests that the results presented here are fairly representative of the SSIZ as a whole, rather than being biased by a particular region or time period. In summary, we have shown that prolonged under ice phytoplankton growth prior to retreat is typical of the Southern Ocean SSIZ.

These findings are broadly in agreement with those presented by Uchida et al. (2019), who analysed the same dataset and found that early growth initiated in August/September in the region south of ∼60°S. However, the authors do not explicitly investigate growth in relation to the release of melt waters in the SSIZ, and appear to conclude that melting generally initiates growth through the release of iron trapped in sea ice, as well as the relief of light limitation.

Our findings are also complementary to those of Briggs et al. (2017), who analysed nine under ice floats deployed in 2014 and 2015 in the Ross and Weddell Seas. Although the authors concluded that respiration dominated during the ice covered period, their Figures 4 and 6 show that production begins before the end of the ice covered period. Indeed, our interest has been the period of initial growth when overall biomass is still generally very low, but growth rates are significant compared to the rest of the growth phase. Briggs et al. (2017) note that nitrate, oxygen and dissolved inorganic carbon (DIC) changes during the ice covered period are consistent with net respiration, however the modest phytoplankton standing stock present at GI (which occurs at the end of the ice covered period) may not be sufficient to appreciably reduce nitrate and DIC concentrations and increase oxygen values (see mean chl-a concentration at GI in Table 2). Thus, the seemingly contradictory conclusions of our results is due to differences in which the period of season the analysis is focussed on, with Briggs et al. (2017) focussing on earlier periods of the year when the respiration signal is dominant, and the work presented here focussing on the early growth period when respiration switches to production. In the end, higher frequency sampling is needed to more precisely determine the timing of net production.

It is also interesting to note that these results can be interpreted as supporting the "disturbance-recovery" hypothesis laid out by Behrenfeld and Boss (2014). Using this framework Behrenfeld and Boss (2014) and Behrenfeld et al. (2017) have argued that growth/bloom initiation occurs much earlier in the year in winter at high latitudes than previously thought, a very similar conclusion to that arrived here. However, as will be discussed below, the winter growth shown here does not necessarily require that ecological interactions be invoked to explain it. Indeed, in our regional box model experiments (discussed below) we found that zooplankton have a lagged response to diatom growth in early spring (see Figure S9), suggesting that other factors are responsible for the timing of initial growth. Furthermore, altering the zooplankton model parameters (such as lowering the diatom availability) did not lead to a phenology resembling the float data. We therefore note that while the role played by ecological interactions is not ruled out here, it is argued that the observed growth can be accounted for by a revision of our understanding of the under ice light environment, as well as the physiological response by phytoplankton.

**Growth Under Extreme Light Limitation**

We now move on to the question of how phytoplankton are able to sustain growth under such poor ambient light conditions. Recall that the average stratification depth at the time of GI is around 130 m, and that satellite data suggest near complete ice cover. Although the timing of GI in October would allow for ample light in open ocean conditions, previous studies suggest that light transmittance through typical consolidated ice would be just 1- 5% of that incident at the surface (even with a thin snow layer, Fritsen et al. (2011)). Two possible explanations for growth under these conditions are then apparent: one, light is more readily available in ice covered environments than previously thought, and two, phytoplankton are more adapted to extreme low light than previously thought. Hence, the phenomenon can be accounted for by both physical factors (such as sea ice and vertical mixing conditions which alter light availability) and biological ones (such as phytoplankton physiology). Both factors are likely operating simultaneously. Indeed, the very presence of growth indicates light levels above zero, suggesting a revision of our current understanding of under ice environments.

In our regional box model experiments we explore both physical and biological factors. The fact winter and spring phenology is brought closer to observations when low light efficiency is enhanced by an order of magnitude (to a value typical of sea ice algae) certainly suggests a role for phytoplankton adaptation (Figure 7 - LLA experiments). However, the interpretation is complicated somewhat by the fact that under certain conditions phenology may be best described by simulations with no ice (OPEN) or with ice but standard physiology (ICE).

For example, in the Weddell Sea (Figure 7A and B) early growth in August is best captured by OPEN experiments, but subsequent spring growth rates (October-November) more closely align with LLA simulations. The inference here would be that in this region sea ice is unconsolidated and highly permeable to light, allowing growth to initiate as soon as incident radiation is sufficient. This corresponds well with the correlation between GI and latitude shown in Figure 4A. This is despite the apparently near 100% sea ice concentration suggested by satellite data (see Figure 6 and Supplementary Figure S7). Indeed, at these latitudes we may actually be in the MIZ, which would explain the higher light permeability. Yet, this is not to say that sea ice has no effect, later in the season growth rates are slowed by its presence, explaining why LLA experiments perform better here. These findings generally agree with previous studies which point to light (as opposed to dissolved iron) being the primary driver of early spring growth in the high latitude Southern Ocean (e.g. see Joy-Warren et al. (2019) and citations therein). We would also note that both silicate and iron are close to their seasonal maximum concentration during late winter/ early spring in all our model experiments (see supplementary Figure S10), thus ruling out nutrient limitation in all regions.

Further south in the Bellingshausen and Ross seas, sea ice is expected to be more consolidated in winter and spring, and so phenology is better captured by LLA simulations (note that the offset in winter time chl-a concentrations seen in these regions in Figure 7C+D is likely due to the need to adjust the metabolic loss terms for phytoplankton in full darkness). However, in two cases the timing of GI most closely matches ICE experiments (see Figure 8, regions B70 and R75). This may be accounted for by especially thick snow and ice layers in those cases, which led to delayed growth. This highlights the importance of the particularities of ice morphological features and their effect on the light environment, something which does not seem to be captured by satellite sea ice concentration.

Thus, it is both the character of ice and snow overhead, and the physiological response to severe light limitation that may address the question raised at the start of this section. A crucial point here is that 100% sea ice cover (in the winter Antarctic sea ice) as seen from satellite does not necessarily imply a completely consolidated ice surface (Vichi et al., 2019). While the ocean may indeed be completely covered, the ice itself may be unconsolidated, being primarily composed of pancakes loosely connected by frazil or brash ice. Such a condition is common in the Southern Ocean, and is maintained by wind and wave action far from the ice edge. Waves are known to propagate several hundred kilometres into the ice, effectively preventing the formation of pack ice-like conditions (Kohout et al., 2014; Meylan et al., 2014). Wind forcing is also known to be highly effective in causing ice break-up and motion, with intense synoptic events in the Weddell and Eastern Indian oceans occurring frequently (Vichi et al., 2019; Uotila et al., 2000). Such events, along with interactions with the westerly wind belt, drive the formation of gaps within the MIZ, as well as within pack ice. Therefore, the highly dynamic nature of Antarctic sea ice may lead to a general enhancement of light availability in the underlying ocean. The presence of even a tiny amount of light

may be expected to induce acclimation in primary producers (that are adapted to low light), thereby explaining why model configurations which take this into account produce a more realistic phenology.

## 5   Conclusions

This study has characterised under ice phytoplankton phenology using a unique dataset of BGC-ARGO profiles, complemented by a set of process-oriented biogeochemical model experiments. We have shown that rather than acting as a trigger as postulated in previous studies, the release of melt waters enhances growth in an already highly active phytoplankton population. This may explain the decline in phytoplankton stocks observed by Veth et al. (1992) in melt water lenses of the north-western Weddell Sea. That is, the decline (in a still highly stratified surface ocean) may be accounted for by the natural reduction occurring in a bloom that already started prior to the melting. Such unexpected early growth (under presumed severe light limitation) may be accounted for by a combination of low light adaptation by phytoplankton and sea ice permeability with respect to light. We argue that such permeability is related to wind and wave forcing, which together preserve an unconsolidated ice morphology that is not captured by current satellite sea ice concentration algorithms.

However, our investigation has not been exhaustive of all possible mechanisms leading to under ice growth. Future research directions could include an examination of potential discrepancies between the timing of shoaling of the mixed layer and that of active turbulent mixing (e.g. Carranza et al. (2018); Sutherland et al. (2014)). An earlier reduction in mixing would increase ambient light, and help explain the observed under ice growth. Other ecological factors could also be explored, such as potential interactions between pelagic and sympagic communities, which are known to be highly efficient at low light intensities (Tedesco and Vichi (2014) and citations therein). Nevertheless, the findings presented here have important implications for our understanding how the biogeochemistry of the region may change in the future. With possible earlier sea ice retreat, as well as a generally thinner and more dynamic ice in some regions (including the Arctic), we may expect even earlier growth then reported here, which would likely alter the seasonal air-sea carbon flux and thus the biological carbon pump.

*Code availability.* Python code used in this study is available at https://github.com/MarkHague/BGC-ARGO-Tools. Specifically, routines used to compute growth initiation (GI) timing and melt onset timing are provided.

*Author contributions.* Mark Hague conducted the float data analysis, performed the model experiments and wrote the manuscript. Marcello Vichi conceptualized the model experiments and provided valuable input and expertise all aspects of presented work.

*Competing interests.* The authors declare no competing interests.

*Acknowledgements.* This research has received funding from the National Research Foundation through the South African National Antarctic Programme (SANAP) and the NRF-STINT bilateral collaboration programme.

Data were collected and made freely available by the Southern Ocean Carbon and Climate Observations and Modeling (SOCCOM) Project

funded by the National Science Foundation, Division of Polar Programs (NSF PLR -1425989), supplemented by NASA, and by the International Argo Program and the NOAA programs that contribute to it. (http://www.argo.ucsd.edu(link is external), http://argo.jcommops.org(link is external)). The Argo Program is part of the Global Ocean Observing System.

We acknowledge the use of the BFM model freely made available by the BFM System Team http://www.bfm-community.eu.

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
