# Peer review of "Southern Ocean BGC-Argo Detect Under Ice Phytoplankton Growth Before Sea Ice Retreat"

_Biogeosciences, 2020_

## Referee Comment (RC1) · Anonymous Referee #1 · 20 Jul 2020

The manuscript examines the "melt-water hypothesis", which hypothesizes that the melting of sea ice in austral spring allows for an increase in photosynthetically available solar radiation (PAR) and triggers a rapid growth in phytoplankton, by using in-situ observations obtained by the biogeochemical (BGC) Argo floats. Although many studies have invoked this hypothesis in explaining the initiation of the spring bloom, they find that the growth initiation (GI) occurs roughly a month before the sea ice starts to melt. They compliment their results with a box model numerical experiment showing that the observed time series can be explained by a combination of physical mechanisms modulating the permeability of the sea ice to PAR and physiological state of the phytoplankton. The paradigm they find where phytoplankton growth is occurring prior

to the melt of sea ice in the seasonal sea-ice zone (SSIZ) is new. I recommend the manuscript for publication with minor revisions listed below.

1. Lines 147-149: Have you looked at whether examining Chl-a below the mixed layer changes the timing of GI? Does the timing of GI differ from the 'onset' proposed by Behrenfeld and Boss (2018)?

2. Lines 175-177: I would like to see how iron and silicate (presumably being the limiting nutrients for diatom) vary over the course of your simulation. Is the ecosystem always light replete or do nutrients also become a limiting factor for growth in your simulation?

3. Lines 178-180: Could you add a figure/time series of the input variables (solar radiation etc.) to the model?

4. Line 220: Thank you for also examining particulate organic matter (POC). Could you add in text how you calculated POC from the BGC-Argo floats? Also, considering the capacity for photoquenching/photoacclimation, do you think POC is a more robust variable in quantifying the temporal variability in phytoplankton/biomass? The weaker dependence on different rates of cooling, as you note, seems to indicate so.

5. Figure 7: Could you comment on the systematic offset between the float data and the ICE/LLA simulations you see in panels C and D?

6. Lines 294-296, 345: The PAR condition seems to be a key factor in interpreting the float and model results. Is it possible to estimate the PAR under sea ice either using satellite data (Morel et al., 2017) or Argo floats? (I am not sure whether the floats you have examined have the instrumental capacity but some BGC-Argo floats measure also PAR.)

7. Lines 324-326: Can you comment on the role/impact of zooplankton in your box model experiments? Does grazing by zooplankton affect the GI timing?

References:

[Figure]

Behrenfeld and Boss. (2018) Student's tutorial on bloom hypotheses in the context of phytoplankton annual cycles. Global change biology, 24(1), 55-77.

Morel et al. (2017). Examining the consistency of products derived from various ocean color sensors in open ocean (case 1) waters in the perspective of multi-sensor approach. Remote sensing of Environment, 111(1), 69-88

––––––––––––––––––––––

---

## Referee Comment (RC2) · Anonymous Referee #1 · 20 Jul 2020

In the second point listed for revision, I meant to say: "light depleted (before and after the timing of GI)" instead of replete, sorry.
* * *

---

## Short Comment (SC1) · 20 Jul 2020

Hi- we in SOCCOM are very glad that you have made such great use of the under ice data from SOCCOM floats. Could we ask that you include our data statement in the acknowledgments? This will help with supporting continued funding from NSF (SOCCOM), NASA (FLBBs on the floats), and NOAA (Argo). We have a statement on our data page https://soccom.princeton.edu/content/data-access

https://soccom.princeton.edu/content/acknowledgment-text

"Data were collected and made freely available by the Southern Ocean Carbon and

[Figure]

Climate Observations and Modeling (SOCCOM) Project funded by the National Science Foundation, Division of Polar Programs (NSF PLR -1425989), supplemented by NASA, and by the International Argo Program and the NOAA programs that contribute to it. (http://www.argo.ucsd.edu, http://argo.jcommops.org). The Argo Program is part of the Global Ocean Observing System."

Thanks very much! Lynne Talley
* * *

---

## Author Comment (AC1) · 28 Jul 2020

Our apologies for not including the full acknowledgement, it will be included in the revised version.

---

## Referee Comment (RC3) · Anonymous Referee #2 · 29 Jul 2020

General Comments:

The authors utilize BGC-Argo float data from the SOCCOM project deployed within the seasonal sea ice zone of the Southern Ocean to estimate the onset of phytoplankton growth in the spring. Large blooms are often observed at the receding ice edge in the Southern Ocean but the initiation of this growth is unclear. One hypothesis is that the ice melt stratifies the upper ocean providing preferential conditions for phytoplankton growth (release of nutrients from ice, increased light, shallow mixed layer depth). Here the authors argue that the initiation of growth begins before the onset of ice melt. They use both temperature and salinity data from the floats to estimate the timing of ice melt

events and in nearly all the floats used in this study, the chl-a signal corresponding to phytoplankton growth, appears before ice melt. The leading hypothesis from this study is that light is the driving force for initiating growth even with >90% ice cover estimated from satellite data. There are many different kinds of sea ice and more light may penetrate through than previously thought. The authors also present a simple model with ice, no ice, and low light adaptation to further assess which driving factors of growth initiation most closely resembled the float data. It was found that the low light adaptation most closely fits the float data providing further support that low levels of light penetrating through the sea ice initiate phytoplankton growth prior to ice melt. I recommend publication of this manuscript with minor revisions listed below.

Specific Comments:

Line 15-21: more references needed

Line 52: define what the marginal ice zone is

Line 71: define date range used rather than all available floats because this will always be changing

In Methods: you need to discuss the sampling frequency of the floats as 10 days earlier on in the methods

Line 92: ice melting already occurred ***or the float moved out of an ice covered region*** (or the ice moved but may not have melted)

Line 173: list the standard nutrients rather than only saying 'all standard nutrients'

Line 174: citation needed or source for the nutrient concentrations used

Line 179: How was MLD defined?

Line 199: Is it more correct to say Bio-Argo? Or is it BGC-Argo? Used both ways in manuscript

Figure 5/6: Captions should be more stand alone and not just refer to the text. I recommend providing a little more detail in the captions.

Line 310: In the Briggs 2017 study a respiration signal was observed in oxygen and DIC inventories during the under ice period but switched to production prior to the estimated ice edge (see figures 4 and 6). There is no clear disagreement between these two studies just a different time-frame focus.

Line 312: Satellite data was also used in this study to estimate ice cover Discussion: Have you compared the MLD at the time of GI for each of the floats?

Technical Corrections:

Abstract: 0D model? Did you mean 0.5D as later referred to in the manuscript?

Line 142: change shown to show

Table 2: inconsistent letter case

All figures: the font is very small. I would recommend increasing all figure text font.

Figure 7: I recommend plotting all four subplots with same size axes. Only the bottom right plot has a larger x-axis.

Line 289: Add 'In' to start of sentence 'Figures 5 and 6...'

Line 370: Needs commas

---

## Author Comment (AC2) · 14 Sep 2020

**Response Reviewer 1**

We would like to sincerely thank the reviewer for their time and valuable input. These comments and suggestions have surely enhanced the quality of the paper.

1. **Lines 147-149: Have you looked at whether examining Chl-a below the mixed layer changes the timing of GI? Does the timing of GI differ from the 'onset' proposed by Behrenfeld and Boss (2018)?**

Part 1 of the Question:

Below we plot the difference in GI when taking into account chl-a below the mixed layer (we average now to ~50 m below the ML by including an additional 7 depth levels below the ML). The left side of the distribution represents events where the timing of GI is moved earlier in the year, the right where GI is delayed. In general, around 45% of events have no change, while ~70% have a change of less than 2 weeks. It is also clear that the overall effect would be to shift the timing of GI earlier in the year, since twice as many events are shifted earlier than are delayed.

These changes appear to driven by the extent to which chl-a concentrations are diluted by including data from below the estimated mixed layer. In cases where GI is shifted earlier, concentrations are significantly reduced, especially in the summer, which then has the effect of shifting the location of the median growth rate earlier. That is, since growth rates are reduced more in late spring/summer (than in the early spring) the median rate now occurs earlier in the year. This is contrasted with cases where GI is delayed, which do not display significant dilution of the spring/summer chl-a concentration (evidently there is some chl-a below the estimated ML). Indeed, in these cases the spring growth rate is enhanced by the inclusion of chl-a below the ML, which shifts the location of the median growth rate later in the season. Another way to think of it is that the late winter growth rates are now smaller compared to those in the spring, so it takes longer for the criteria for GI to be met. We note that this only delays GI by more than 2 weeks in 4 cases, so the presence of significant chl-a below the ML appears to be fairly rare in this data set. We will add this point to the revised manuscript at line 216.

[Figure]

Our definition of GI is distinct from that proposed by Behrenfeld and Boss (2018). Firstly, in their paper they refer to "bloom initiation" (their table 1), where as our definition of "growth initiation" is intentionally indifferent to whether or not a "bloom" occurs after GI, for reasons that are discussed in lines 156 - 162. Second, the Behrenfeld and Boss (2018) definition requires that the loss term be estimated, where as we define GI purely based on the time derivative of mixed layer chl-a.

2. **Lines 175-177: I would like to see how iron and silicate (presumably being the limiting nutrients for diatom) vary over the course of your simulation. Is the ecosystem always light depleted or do nutrients also become a limiting factor for growth in your simulation?**

Below we plot time series of iron (brown) and silicate (green) in each of the 4 study regions (only for the "LLA" experiment). The shaded regions correspond to the variability present in each study region (both spatial and temporal) as is discussed in the paper. Neither nutrients are limiting for the late winter/early spring period under consideration. For completeness we will add this figure to the supplementary material and add this point to the revised manuscript.

[Figure]

**3.  Lines 178-180: Could you add a figure/time series of the input variables (solar radiation etc.) to the model?**

We thank the reviewer for bringing this up and will add the figure below to the supplementary material. Note that the sea ice concentration forcing is already shown in the Supplementary figure S5 for the Weddell and Bellingshausen Sea regions, and for the Ross Sea it is shown in the main paper in Figure 6. One detail which was left out of the paper is that the more northerly experiments in the Weddell Sea (W60 and W65) used an analytical light forcing. This will now be added to section 2 of the paper.

[Figure]

**4.  Line 220: Thank you for also examining particulate organic matter (POC). Could you add in text how you calculated POC from the BGC-Argo floats? Also, considering the capacity for photoquenching/photoacclimation, do you think POC is a more robust variable in quantifying the temporal variability in phytoplankton/biomass? The weaker dependence on different rates of cooling, as you note, seems to indicate so.**

POC comes as a variable in the SOCCOM float files. More information regarding its calculation can be found in Boss & Haëntjens (2016), although this equation is given:

POC = 3.23 × 104 × bbp(700) + 2.76 [mg m–3 ], where bbp(700) refers to particle backscattering at 700nm measured by the float.

In terms of which quantity is a more robust estimate, we would argue that both are associated with uncertainties of a similar magnitude (as is discussed in Boss & Haëntjens, 2016). We agree that POC appears to be a more stable estimate of the amount of scattering particles under sea ice. However, the relationship between backscattering and phytoplankton carbon is still quite uncertain in the Southern Ocean, as has been analysed by Thomalla et al. (2017). Once strengthened with additional studies, the relationship between POC and chlorophyll would certainly help to better understand the under ice acclimation. With regard to quenching, we believe that it is unlikely to play a role under sea ice, given the low light levels.

5. **Figure 7: Could you comment on the systematic offset between the float data and the ICE/LLA simulations you see in panels C and D?**

We assume the reviewer refers to the offset in the winter time chl-a concentrations?
The model was not specifically tuned to simulate the transitional periods between autumn winter and spring-summer. As explained in the text, the model was meant to explore the mechanisms during the melting phase, and therefore the focus was on the relative growth rates and the phenology. The discrepancy is likely due to the need to adjust the metabolic loss terms for phytoplankton in full darkness, a point which will be added to the discussion in section 4.

6. **Lines 294-296, 345: The PAR condition seems to be a key factor in interpreting the float and model results. Is it possible to estimate the PAR under sea ice either using satellite data (Morel et al., 2017) or Argo floats? (I am not sure whether the floats you have examined have the instrumental capacity but some BGC-Argo floats measure also PAR.)**

The float data we have used unfortunately do not include any PAR information. Looking at the global BGC-Argo array and filtering for irradiance data reveals that under ice data are exceedingly rare in the Southern Ocean (2, possibly 3 floats may have some profiles under ice). Rather, under ice floats with that capability have exclusively been deployed in the North Atlantic. This point will be added to section 2 at line 66. It may be possible to infer the under ice light environment in the Southern Ocean based on data in the North Atlantic (if one could find profiles which sampled under ice conditions similar to that found in the Southern Ocean - i.e. fairly thin and unconsolidated ice).

In terms of satellite data, as far as we know there is no way to estimate irradiance under sea ice from space. In open ocean regions one may estimate the diffuse attenuation coefficient and thereby infer the depth of the euphotic zone (as is done in Morel et al., 2017). While it may be possible to estimate this quantity in rare cases where an observation is retrieved from an open ocean region within the SSIZ (e.g. leads), this would not address the question of how light is

transmitted through thin and frazil (ice/water mixture) ice. We say that such observations would be rare because ocean colour data are generally missing in the winter/early spring months south of 60S under consideration here.

**7. Lines 324-326: Can you comment on the role/impact of zooplankton in your box model experiments? Does grazing by zooplankton affect the GI timing?**

This is an important point and we thank the reviewer for highlighting it. In the model, zooplankton does not seem to affect the phenology of the diatoms in any of the three core experiments (OPEN, ICE or LLA). Rather, zooplankton have a lagged response to the diatom growth in early spring, suggesting that other factors are responsible for the timing of initial growth. This relationship is clearly shown in the figures below, which plot time series of diatom and mesozooplankton concentrations for each study region for the "LLA" experiment. The other 2 core experiments show the same relationship.

However, we are not making the argument that zooplankton play no role in phenology in the real Southern Ocean, just that the under ice growth identified in the float data can be accounted for without assuming a strong role for zooplankton. Furthermore, altering the zooplankton model parameters (such as lowering the diatom availability) did not lead to a phenology resembling the float data. Again, we will add this figure to the supplementary material and add some discussion of these points at lines 324 - 326.

[Figure]

References:

Boss, E.B. and N. Haëntjens, 2016. Primer regarding measurements of chlorophyll fluorescence and the backscattering coefficient with WETLabs FLBB on profiling floats. SOCCOM Tech. Rep. 2016-1. http://soccom.princeton.edu/sites/default/files/files/SOCCOM_2016-1_Bio-opticsprimer.pdf.

Thomalla, S. J., Ogunkoya, A. G., Vichi, M. and Swart, S. 2017.  Using Optical Sensors on Gliders to Estimate Phytoplankton Carbon Concentrations and Chlorophyll-to-Carbon Ratios in the Southern Ocean. *Frontiers in Marine Science.* 4, 34.

---

## Author Comment (AC3) · 5 Oct 2020

**Response Reviewer 2**

We would like to sincerely thank the reviewer for their time and valuable input. The suggested changes will surely enhance the quality of the paper.

**1. Line 15-21: more references needed**

More references have been added to these lines, including Sallee *et al.,* 2015*,* Ardyna *et al., 2017* and Hague & Vichi, 2018. References for missing data are Cole *et al.,* 2012 and Racault *et al.,* 2012.

**2. Line 52: define what the marginal ice zone is**

While we do offer a definition in the subsequent line we agree that this definition is not specific enough. The following clarification has been added to the revised manuscript:

" *Note that we would define the MIZ here by dynamical considerations such as wave propagation (i.e. the MIZ may be defined as the region where wave attenuation is below a given threshold) and not a satellite ice concentration threshold (for example, see Squire et al., 2007; Meylan et al., 2014).* "

**3. Line 71: define date range used rather than all available floats because this will always be changing**

We thank the reviewer for raising this point and will update the manuscript. We have also included a table to the supplementary material which contains information (float ID, year, location) for all the floats analysed in the paper, which we show below.

**Table 1.** Table of all floats used in this study, including both the WMO ID and MBARI ID (for identifying floats on the SOCCOM website). The years of data which where used are shown, although there may be more available data at the time of reading. Mean locations for each float are also shown based on the time interval used for calculations.

| WMO ID / MBARI ID | Years Sampled | Mean Latitude | Mean Longitude |
|---|---|---|---|
| 5904768 / 0570SOOCN | 2016 | 64.8°S | 166.7°W |
| 5904671 / 0507SOOCN | 2016 2017 2018 | 62.3°S | 82°E |
| 5905636 / 12754SOOC | 2018 | 67°S | 149.4°W |
| 5905078 / 12371SOOC | 2017 | 66.6°S | 124.4°W |
| 5904858 / 12551SOOC | 2017 2018 | 73.9°S | 148.7°W |
| 5904184 / 9091SOOCN | 2014 | 61.4°S | 147.4°W |
| 5904469 / 9096SOOCN | 2018 | 60.4°S | 23.2°E |
| 5904767 / 0568SOOCN | 2016 2018 | 63.2°S | 145.5°W |
| 5904859 / 12549SOOC | 2017 2018 | 70.2°S | 104.3°W |
| 5904857 / 12381SOOC | 2017 2018 | 72.8°S | 167.2°W |
| 5905100 / 12361SOOC | 2017 2018 | 65.2°S | 166.3°E |
| 5904860 / 12541SOOC | 2017 2018 | 72.1°S | 164.8°W |
| 5905075 / 8501SOOCN | 2017 2018 | 68.9°S | 102.8°W |
| 5904472 / 9275SOOCN | 2015 2016 2017 2018 | 68.5°S | 25.9°W |
| 5904855 / 12559SOOC | 2017 2018 | 68.3°S | 83.8°W |
| 5905077 / 12379SOOC | 2017 | 65.7°S | 107.1°W |
| 5904468 / 9099SOOCN | 2015 2016 2017 2018 | 65°S | 2.3°E |
| 5904471 / 9094SOOCN | 2015 2016 2017 2018 | 65.7°S | 4°E |
| 5904397 / 9125SOOCN | 2015 2016 2017 2018 | 61.2°S | 2.4°W |
| 5905080 / 12366SOOC | 2017 | 65°S | 117.7°W |

**4. In Methods: you need to discuss the sampling frequency of the floats as 10 days earlier on in the methods.**

This has been added at line 72:

"*Analysis was done on data from 2014 - 2019, making use of chl-a, pressure, temperature, salinity and position data available at a 10-day frequency.* "

**5. Line 92: ice melting already occurred \*\*\*or the float moved out of an ice covered region\*\*\* (or the ice moved but may not have melted)**

This is a very good point, we have added these additional possibilities.

**6. Line 173: list the standard nutrients rather than only saying 'all standard nutrients' + Line 174: citation needed or source for the nutrient concentrations used.**

All nutrients are now added at line 173, along with citations for concentrations:

" *In terms of nutrients, phosphate, nitrate and ammonium are included, as well as silicate and iron. Initial nutrient conditions where chosen to be representative of the Southern Ocean south of ~60°S, with non-limiting concentrations of nitrate (31.8 mmol/m$^3$), phosphate (2 mmol/m$^3$) and silicate (40 mmol/m$^3$) (Sarmiento and Gruber, 2006). An initial dissolved iron concentration of 0.3 μmol/m$^3$ (Tagliabue et al., 2014) is applied to all experiments, which gave the most realistic magnitude of summer growth when compared to float data.* "

**7. Line 179: How was MLD defined?**

This is an oversight on our part, thank you for pointing it out. The MLD is defined as the depth at which the Brunt-Viasala frequency is maximal in the water column, which is mentioned later in the text under "Melt Detection." We have added text pointing the reader to this definition at the first mention of MLD at line 179.

**8. Line 199: Is it more correct to say Bio-Argo? Or is it BGC-Argo? Used both ways in manuscript.**

We will change all references to BGC-ARGO.

**9. Figure 5/6: Captions should be more stand alone and not just refer to the text. I recommend providing a little more detail in the captions.**

Captions have been updated in the revised manuscript as follows:

"*Figure 5: Distribution of the difference in timing (in days) between growth initiation (GI) and melt onset (for all floats sampling under ice). GI is defined as the point at which the time derivative of mean mixed layer chl-a exceeds the median time derivative (computed for the growth period).*

*Negative values in the distribution indicate that GI has occurred prior to the detected melt onset. Curved lines represent the probability density functions for several values of the assumed cooling threshold (rc) in the upper ~20 m of the water column. This value represents an assumed decrease in temperature over the upper ~20 m, which is required to delineate under ice from open ocean profiles (since floats do not sample the upper ~20 m in winter, they do not sample water below the freezing point).* "

" *Figure 6: Satellite sea ice concentration (SIC) versus ARGO-float chl-a for the region R75. Shaded regions around each line represent both the spatial and temporal variability present in each dataset.That is, each bold line plots the mean value of 5 time series which are associated with a specific melt event. Events are separated in space and time; in this particular case 2017 and 2018 were sampled by 3 floats (see table 1), which resulted in 5 time series (2 each, with one of the floats only sampling in 2017). The red star represents the mean value of GI.* "

**10. Line 310: In the Briggs 2017 study a respiration signal was observed in oxygen and DIC inventories during the under ice period but switched to production prior to the estimated ice edge (see figures 4 and 6). There is no clear disagreement between these two studies just a different time-frame focus.**

We thank the reviewer for bringing this point to our attention and agree that the studies are focussed on different periods. We would also add that Briggs *et al.* 2017 is also more focussed on blooms (i.e. high chl-a concentration, as they compare years with and without a bloom in their Figure 3), while our study is more focussed on growth in general (i.e. relative changes in chl-a as opposed to absolute magnitudes). The manuscript has been updated to reflect these points, with the following statement added within the paragraph starting at line 310:

"*Thus, the seemingly contradictory conclusions of our results is due to differences in which period of season the analysis is focussed on, with Briggs et al. (2017) focussing on earlier periods of the year when the respiration signal is dominant, and the work presented here focussing on the early spring period when respiration switches to production.*"

**11. Line 312: Satellite data was also used in this study to estimate ice cover Discussion: Have you compared the MLD at the time of GI for each of the floats?**

At lines 220-221 we state that the average stratification depth (our metric of MLD) at GI is 129 m. We have now added that the standard deviation is 51 m, highlighting fairly substantial variability present in the data set. In addition, we have added a figure showing the relationship between stratification depth and GI to the supplementary material, which we show below. We have added to the manuscript at line 221 that there is no correlation between the extent of vertical mixing (i.e. stratification depth) and GI, as well as commented on the range of values found for the stratification depth.

[Figure]

**Figure 1:** Stratification Depth (Nd) at the timing of GI plotted against GI for each of the 42 melt events detected. Overlain in blue is the linear regression with the 95% confidence intervals for 1000 bootstrapped resamples shaded in light blue. Histograms and PDFs of each variable are shown along the edge of the axes.

**Technical Corrections**

**12. Abstract: 0D model? Did you mean 0.5D as later referred to in the manuscript?**

Yes, we were referring to the 0.5D model presented in the study. We felt that specifying 0.5D in the abstract would lead to potential confusion, since most readers would be familiar with 0D models but not 0.5D. Therefore, we have updated the abstract to read "box model with varying vertical depth" as follows:

*"This led to the development of several box model experiments (with varying vertical depth) in which we sought to investigate the mechanisms responsible for such early growth."*

**13. Line 142: change shown to show**

This has been changed.

**14. Table 2: inconsistent letter case**

Letter case is consistent now.

**15. All figures: the font is very small. I would recommend increasing all figure text font.**

We have increased the font of all figures.

**16. Figure 7: I recommend plotting all four subplots with same size axes. Only the bottom right plot has a larger x-axis.**

We assume the reviewer is referring to the y-axis, as the x-axes are all the same size. We have updated the figure as suggested.

**17. Line 289: Add 'In' to start of sentence 'Figures 5 and 6…'**

Typo corrected.

**18. Line 370: Needs commas**

We would argue that adding commas changes the meaning of the sentence in a way that is not intended. However, we agree that the sentence needs to be changed to be more understandable and therefore suggest the following:

*"The presence of even a tiny amount of light may be expected to induce acclimation in primary producers (that are adapted to low light), thereby explaining why model configurations which take this into account produce a more realistic phenology. "*

**References**

Ardyna, M., Claustre, H., D'Ortenzio, F., van Dijken, G., Arrigo, K. R., D'Ovidio, F., Gentili, B., and Sallée, J.B.: Delineating environmental control of phytoplankton biomass and phenology in the Southern Ocean, Geophysical Research Letters, 44, 5016–5024, https://doi.org/10.1002/2016gl072428, 2017.

Cole, H., Henson, S., Martin, A., and Yool, A.: Mind the gap: The impact of missing data on the calculation of phytoplankton phenology metrics, Journal of Geophysical Research: Oceans, 117, https://doi.org/10.1029/2012JC008249, https://agupubs.onlinelibrary.wiley.com/ doi/abs/ 10.1029/2012JC008249, 2012.

Hague, M. and Vichi, M.: A Link Between CMIP5 Phytoplankton Phenology and Sea Ice in the Atlantic Southern Ocean, Geophysical Research Letters, 45, https://doi.org/10.1029/2018GL078061, 2018.

Racault, M. F., Le Quéré, C., Buitenhuis, E., Sathyendranath, S., and Platt, T.: Phytoplankton phenology in the global ocean, Ecological Indicators, 14, 152–163, https://doi.org/10.1016/j.ecolind.2011.07.010, http://dx.doi.org/10.1016/j.ecolind.2011.07.010, 2012.

Sallée, J.-B., Llort, J., Tagliabue, A., and Levy, M.: Characterization of distinct bloom phenology regimes in the Southern Ocean, ICES Journal of Marine Science, 72, 1985 – 1998, https://doi.org/10.1038/278097a0, 2015.

---

## Author Comment (AC4) · 12 Oct 2020

This point has been addressed in the full response. See the attached pdf in "Response to Reviewer 1."